# Reviewing PTBP1 Domain Modularity in the Pre-Genomic Era: A Foundation to Guide the Next Generation of Exploring PTBP1 Structure–Function Relationships

**DOI:** 10.3390/ijms241311218

**Published:** 2023-07-07

**Authors:** Christine Carico, William J. Placzek

**Affiliations:** Department of Biochemistry and Molecular Genetics, University of Alabama at Birmingham, Birmingham, AL 35294, USA

**Keywords:** polypyrimidine tract binding protein 1 (PTBP1), RNA recognition motif (RRM), ribonucleoprotein 1 (RNP1), ribonucleoprotein 2 (RNP2), RNA, structure–function

## Abstract

Polypyrimidine tract binding protein 1 (PTBP1) is one of the most well-described RNA binding proteins, known initially for its role as a splicing repressor before later studies revealed its numerous roles in RNA maturation, stability, and translation. While PTBP1’s various biological roles have been well-described, it remains unclear how its four RNA recognition motif (RRM) domains coordinate these functions. The early PTBP1 literature saw extensive effort placed in detailing structures of each of PTBP1’s RRMs, as well as their individual RNA sequence and structure preferences. However, limitations in high-throughput and high-resolution genomic approaches (i.e., next-generation sequencing had not yet been developed) precluded the functional translation of these findings into a mechanistic understanding of each RRM’s contribution to overall PTBP1 function. With the emergence of new technologies, it is now feasible to begin elucidating the individual contributions of each RRM to PTBP1 biological functions. Here, we review all the known literature describing the apo and RNA bound structures of each of PTBP1’s RRMs, as well as the emerging literature describing the dependence of specific RNA processing events on individual RRM domains. Our goal is to provide a framework of the structure–function context upon which to facilitate the interpretation of future studies interrogating the dynamics of PTBP1 function.

## 1. Introduction

While it is now known as a ubiquitous regulator of RNA biogenesis, polypyrimidine tract binding protein 1 (PTBP1)—also known as hnRNP I (heterogeneous nuclear ribonucleoprotein 1) [1]—was initially identified as a sequence-specific splicing factor [1,2,3,4,5]. Early dogma in splicing biology suggested that the 5′ splice site of RNA transcripts is recognized solely by sequence complementarity with U1 snRNA (small nuclear RNA) [6]; however, later studies suggested that other sequence-specific proteins were required for accurate and precise 5′ splice site selection [7,8]. PTBP1 was one of these identified proteins. It was found to be a component of the spliceosome that binds pre-mRNAs specifically at the polypyrimidine tract, and its binding affinity directly correlated to splicing efficiency [1,2,3,4,5,9]. PTBP1’s role in alternative splicing was well characterized for many gene targets [4,9,10,11,12,13,14,15,16,17] before its critical roles in other biological processes (e.g., polyadenylation [18], mRNA stability and transport [19,20,21], internal ribosome entry site (IRES)-mediated translation [22,23,24]) were later described, firmly establishing this protein as a central regulator of RNA biogenesis. One well-studied example is the PTBP1 splicing of its homolog, polypyrimidine tract binding protein 2 (PTBP2), that suppresses PTBP2 expression. In specific tissues, downregulation of PTBP1 occurs during differentiation, and causes the expression of the function PTBP2 mRNA and, ultimately, a PTBP2 protein that modulates a distinct profile of RNA regulation [11]. However, while these biological roles have been well described, the mechanisms by which PTBP1 coordinates these diverse functions for a broad pool of RNA targets are still poorly understood.

Structurally, PTBP1 is comprised of an N-terminal bipartite nuclear localization and export signal (NLS/NES) [25,26,27,28] and four RNA binding domains (RBDs) of the RNA recognition motif (RRM) variety (designated RRM1-4) [29,30,31,32,33,34]. There are several types of RNA binding domains (e.g., K homology (KH) domain, zinc-finger domain), but the RRM is the most abundant and well-characterized, both structurally and biochemically [29,35,36,37]. Structural modularity is a common theme in RNA binding proteins (RBPs), as many of these proteins, like PTBP1, are comprised of multiple repeats of a few defined protein domains (e.g., KH domain, RRM, etc.) [37]. For many of these RBPs, studies have shown that these repeating domains can either have an independent function, or cooperate with one another to coordinate the ultimate biological function of the protein. For example, the identity and organization of domains in an RBD can increase RNA binding affinity through additive affinity [38], or create an extended RNA recognition interface via interaction of several domains [39,40,41,42,43]. Additionally, individual domains can act as tethers for adjacent domains, as short linker regions can increase the effective local concentration of a neighboring domain to augment functional affinity [44]. Finally, RNA binding domains (especially RRMs) can serve as protein–protein interaction interfaces that have important impacts on downstream protein function [45,46,47].

## 2. The RRM Domains of PTBP1 Are Structurally Heterogenous

As mentioned above, PTBP1 contains four RNA recognition motifs. Canonically, an RRM contains approximately 90 amino acids arranged in an αβ-sandwich with β1α1β2β3α2β4 topology [29,35], in which the four anti-parallel β-strands form a β-sheet that is packed against the two α-helices [29,35,48,49]. RNA binding takes place on the β-sheet, and is coordinated by a series of intermolecular interactions (e.g., pi-stacking, hydrophobic, hydrogen-bonds, electrostatic, etc.) [29,35,37] between solvent exposed residues and target RNA. Within the β-sheet, there are two conserved RNA recognition sequences, termed RNP1 (ribonucleoprotein 1) on β3 (canonical sequence: [RK]-G-[FY]-[GA]-[FY]-[ILV]-X-[FY]), and RNP2 (ribonucleoprotein 2) on β1 (canonical sequence: [ILV]-[FY]-[ILV]-X-N-L), that are positioned to directly interact with target RNA (Table 1) [29,36,50,51]. In the protein–RNA interface, the RNA positions two nitrogenous bases to stack with conserved aromatic rings at position 5 of RNP1 and position 2 of RNP2 (Table 1). In addition, the RNA sugar rings contact a hydrophobic side chain at position 3 of RNP1 (also a conserved aromatic residue), and the phosphodiester group electrostatically interacts with a positively charged residue at position 1 of RNP1 (Table 1). Of note, although RRMs contain these conserved sequences, most do not make all four of these canonical contacts with target RNA dinucleotides, and typically only make between one and three of these interactions. RNA target specificity is further mediated by intermolecular interactions (e.g., hydrophobic interactions, hydrogen bonds, etc.) between target RNA and residues outside of these conserved sequences (especially on the β4 and β2 strands, as well as the N- and C-terminal regions), similar to an induced fit ligand-protein interaction.

PTBP1’s four RRMs are arranged in a “beads on a string” orientation, where each RRM is joined to its neighbor by linker regions of variable length. The linker regions following RRM1 (42 residues) and RRM2 (58–84 residues) are long, and these domains have been shown to move independently of their neighbor in solution [1,5,30,33,52,53]. However, the linker region between RRM3 and RRM4 is short (24 residues [31,53]), and these domains interact with one another via their α-helical interface to form an intramolecular didomain complex with both β-sheets facing opposing directions [32,33]. This organization of tandem RRMs is unusual, as only a handful of RNA binding proteins contain RRMs that demonstrate stable intramolecular contacts [33,54,55,56,57,58,59], and even fewer make these stable contacts via their respective α-helices (hnRNP A1 [60,61], hnRNP L [59], PTBP1/PTBP2 [33]). Interestingly, the RRMs of PTBP1 also demonstrate significant divergence from the canonical RRM primary sequence and topologic organization [1,5,35,36]. Specifically, in each RRM of PTBP1, key RNA-interacting aromatic residues in the consensus RNP1 and RNP2 sequences are often replaced by hydrophobic residues (discussed in detail below) [1,36,62].

PTBP1 was initially described to bind short, single-stranded pyrimidine sequences (such as UCUU, CUCUCU), including those found at splice junctions [10,11,63,64], and later structural interrogation using short oligos determined that each individual RRM demonstrates sequence context preferences using short oligos (in terms of sequence length, composition, and secondary structure). Here, we review the apo structure of each RRM of PTBP1, and the key residues involved in RRM: RNA interactions, and sequence and structure preferences of target RNA for each RRM. As RRM3 and RRM4 invariably exist as a didomain complex, they will be discussed together (and referred to as RRM3-4). We would like to note that there is extensive evidence that multiple RRMs (and even PTBP1 proteins) are involved in interactions with target RNA in the in vivo setting; however, to our knowledge, there are no detailed structural models of the full PTBP1 protein complexed with RNA. Therefore, the information summarized below reflects RRM: RNA interactions between each RRM (or RRM complex in the case of RRM3-4) and its minimal binding register.

### 2.1. RNA Recognition Motif 1 (RRM1)

RRM1 of PTBP1 demonstrates the canonical RRM topology of β1α1β2β3α2β4 [30,65]; however, as introduced above, its RNP1 and RNP2 sequences reflect several notable deviations from the consensus sequence, namely the substitution of key aromatic residues [36,66] (Table 1, Figure 1a). Early work to characterize the structure of RRM1 bound to RNA revealed that RRM1, as was observed for the full length PTBP1 protein, binds short pyrimidine stretches [30,31,53,67]. Utilizing a short pyrimidine sequence (C1U2C3U4 [31,53]), specific RRM contacts with RNA were mapped, identifying the minimal binding register and mode by which it is coordinated on the RRM1 surface. RRM1 binds U2C3U4 and, consistent with canonical RRM:RNA interactions, RRM1 binds the C3 nucleotide by a pi-stacking interaction with H62 [68] (position 2 of RNP2). Notably, this H62 is a non-canonical substitution of the conserved aromatic residue of RNP2 at position 2 (Table 1); however, its planar aromatic ring can engage in stacking interactions with nitrogenous bases, thereby retaining the functional properties of this conserved position in RNP2. This interaction is further stabilized by a hydrogen bond with the main chain of N132 [68,69], and side chains of F130 and S131 [31,53,69]. These residues are located on β4 outside of the consensus RNP1 and 2 sequences and, thus, are likely contributors to the induced fit specificity of this interaction. At the 3′ end of this C3 nucleotide, the U2 nucleotide sits above β4, is H-bonded to the side chain of Q129 (via its O2), and stacks with the side chain of R64 (position 4 of RNP2). At the 5′ end of the C3 nucleotide, the U4 nucleotide sits above β2 in a hydrophobic pocket formed by five protein side chains: L136 [68,69] (C terminus), H133 [68] (β4), F98 (β3—RNP1 position 5), L91 [68] (β2), and L89 [68] (β2) [30,31] (Figure 1b). Importantly, subsequent studies have confirmed the involvement of many of these residues in binding RNA with different structural context (pyrimidine sequence in a loop region of IRES elements), and these are referenced above [68,69]. These studies identified additional RRM: RNA contacts, and it is thought that this is due to the presence of multiple distinct binding registers, as well as the structure of the RNA itself. Of note, of the two conserved aromatic residues in the consensus RNP1 sequence, the F98 residue at position 5 is the only retained aromatic, although rather than engaging in a stacking interaction as in most other RRMs, it contributes to hydrophobic pocket formation and, thus, the induced fit specificity of this interaction (Table 1).

Based on these data, the preferred binding register for RRM1 is YCU with only the C nucleotide engaging in a canonically described interaction with conserved features of the RNA binding interface. The first position of this binding register can be occupied by either pyrimidine (C or U), because Q129 on β4 can act as either a hydrogen bond donor or acceptor, and can thus accommodate either pyrimidine [31]. Of note, although these structural studies mapped RRM:RNA contacts with a single-stranded tetranucleotide, several studies have demonstrated that RRM1 (and RRM2, as it was studied as a PTBP1 subdomain containing both RRM1 and RRM2) preferentially binds loop structures [67,70,71].

Intriguingly, several recent studies have demonstrated that the C-terminal region of the RRM1 domain is a critical allosteric regulator of RNA binding [68,72]. These studies identified a C-terminal α-helix (termed the α3 helix) comprised of residues 144–154 that folds upon binding to the stem-loop region of the encephalomyocarditis virus (ECMV) IRES element, but does not directly interact with RNA. This α3 helix is thought to serve as sensor of RNA secondary structure, and acts as an allosteric regulator of RNA binding—a phenomenon seen in other RRM-containing proteins [39,40,41,42]—suggesting that the C-terminal region of RRM1 is not simply displaced, but plays a significant role in allosterically regulating RNA binding in a structure-dependent context. This provides significant context for the studies that demonstrated that RRM1 preferentially binds loop structures [67,70,71].

### 2.2. RNA Recognition Motif 2 (RRM2)

Unlike RRM1, RRM2 demonstrates an extension of the canonical RRM topology with an additional fifth β-strand that sits adjacent and anti-parallel to β2, stabilized by a stacking interaction between Y275 (β5) and H201 (α1 helix) [30,73]. β4 and β5 are connected by a long loop region (13 residues) that sits in a hydrophobic pocket created by several residues on the β-sheet interface (Figure 2a) [30]. Additionally, residues Y267-N269 are arranged as in a “pseudo-sixth strand” configuration anti-parallel to β5, with residues beyond N269 folding back to contact the β4- β5 loop [30] (Figure 2a). This β1α1β2β3α2β4β5 topology with an additional pseudo-sixth β-strand extends the β-sheet RNA binding interface, as compared to canonical RRMs. In addition, the β1, β3 and β4 strands are longer than in most RRM domains; taken together, these modifications significantly expand the available RNA binding interface [30].

Studies characterizing RRM2′s interactions with various short pyrimidine sequences (C1U2C3U4, C1U2C3U4C5, C1U2C3U4C5U6) identified that RRM2 binds the C3U4 doublet, as well as U6 [31,53]. As with RRM1, the C3 nucleotide is sandwiched between two sidechains: R185 on β3, and K259 on β4-β5 loop. Intriguingly, while R185 is not part of RNP2, sequence alignment reveals that the R185 residue sits in the location that is canonically occupied by position 2 of RNP2, as the conserved RNP2 sequence of RRM2 is shifted by two positions second to an insertion in the β1 strand. Ultimately, although the functional properties of the residue and mechanism of interacting with target RNA is not retained, the position of the interacting residue is conserved (Figure 2a). As seen in RRM1, the U4 nucleotide is bound in a hydrophobic pocket (created by side chains I214 (β2), F216 (β2), L225 (β3—RNP1 position 5), L260 and L263 (both in the β4-β5 loop), with both bases being sequence-specifically recognized by both the main chain of the protein and by S258 (Figure 2b). A subsequent study demonstrated that mutation of the I214 and F216 (as well as K218) residues robustly ablated binding to target sequence within the ECMV and poliovirus-1 (PV-1) IRESs, confirming the role of these residues in RNA binding, even in different structural contexts [69]. L225 is at position 5 of RNP1, which canonically contains an aromatic residue that stacks with RNA bases. However, as seen in RRM1, the residue at this position contributes to hydrophobic pocket formation rather than aromatic stacking. RRM2 does not contact the immediately adjacent nucleotide (C5) and, instead, the U6 nucleotide is in contact with K266, Y267 (located immediately adjacent to and in the pseudo-sixth strand, respectively), and K271 (which beyond the pseudo-sixth strand proximal to β5) (Figure 2b) [31,53].

Based on these data, the minimal binding register of RRM2 is CU(N)N, which is extended by an additional nucleotide as compared to the minimal binding register for RRM1 due to the U6 interaction with the C-terminal extension of this domain (the β5 and pseudo-sixth strand) not seen in RRM1. An additional study using longer sequences than the minimal binding register has also confirmed that there are significant chemical shift perturbations upon RNA binding in the β4-β5 loop, the β5 strand, and the pseudo-sixth strand, as reported above [30]. Although these structural studies demonstrate that RRM2 canonically binds single-stranded RNA, as with RRM1, subsequent studies have suggested RRM2 preferentially binds pyrimidine-rich sequences in loops and bulges [67,70,71,74].

### 2.3. RRM2-RRM3 Linker

The linker region C-terminal to RRM2 (between RRM2 and RRM3) has important structural considerations. Notably, this linker region appears to be arranged in a relatively compact globular conformation, as compared to the RRM1-RRM2 linker, such that RRM2 and RRM3 are in closer proximity than RRM2 and its N-terminally adjacent domain RRM1 [52]. Alternative 3′ splice site selection at exon 9 of this linker region produces three isoforms of PTBP1 (listed from shortest to longest): PTB1, PTB2, and PTB4 [1,5,75]. PTB1 is the shortest isoform, with exon 9 completely excluded from the final transcript, whereas PTB2 contains a truncated exon 9 (and thus an additional 19 residues in this linker region), and PTB4 contains the full exon 9 (an additional 26 residues in this linker region) [76]. This difference in RRM2–RRM3 linker length appears to have important biological implications, as these isoforms have cell-type specific expression [77], and have been shown to have differential splicing repression activity on α-tropomyosin exon 3 (but not α-actinin, another known PTBP1 target) and IRES-driven translation of human rhinovirus-2 [78,79,80]. The differing biological enrichments and activities of these PTBP1 isoforms that differ only at this linker region indicates that its length and conformation have important functional consequences.

### 2.4. RNA Recognition Motifs 3 and 4 (RRM3-4)

Note that all residue numbers correspond to the longest isoform (PTB-4) that contains the entirety of exon 9 (and, thus, the full linker region between RRM2 and RRM3). RRM3 and RRM4 of PTBP1 (RRM3-4), like their N-terminal neighbors, demonstrate several non-canonical modifications. RRM3, like RRM2, contains a β5 strand that sits antiparallel to β2, and is connected to β4 by a long loop that is positioned above the RNA binding interface. RRM4, like RRM1, contains the canonical RRM topology, with four anti-parallel β-strands comprising the β-sheet [31,32,33]. However, unlike RRM1 and RRM2, which are followed at their C-termini by long linker regions that allow their independent movement in solution, RRM3-4domains are separated by a short linker region of only 25 residues. There are extensive interdomain contacts between the α-helical interfaces of each domain that positions them with a fixed orientation relative to one another, in which their β-sheets point in opposing directions (Figure 3). This large interdomain interface is largely hydrophobic, and involves both RRM–RRM and RRM–linker contacts. The α2 helix of RRM4 is positioned perpendicularly to the α1 helix of RRM3, and interacts with the α2-β4 loop of RRM3 (see Figure 3 for specific residue interactions). A large portion of the interdomain linker contributes to interdomain interactions, and a majority of these are with α1 and α2 of RRM3, with two additional contacts with α2 and β4 of RRM4 (see Figure 3 for specific residues). While these interdomain contacts (both between RRMs and between the linker region and either RRM) are largely hydrophobic, the interaction is also stabilized by an ion pair (K424 on RRM3 and E528 on RRM4) [33,66] (Figure 3). Notably, mutation of three side chain residues on the α2 helix of RRM4 (E528, V531 and I535) was sufficient to ablate this interdomain interaction, indicating that this region of RRM4 is particularly critical in forming the interdomain interface [31].

Due to the heterodimeric form adopted by RRM3-4, RRM: RNA interactions for the individual RRM3 and RRM4 domains were determined using the RRM3-4 didomain, in order to provide a more accurate structural interrogation of RNA interactions. As with RRM1 and RRM2, short pyrimidine sequences (C1U2C3U4C5U6) were utilized to identify the minimal binding register of each tandem domain and map specific interactions [31,53]. RRM3 interacts with the U2C3U4C5U6 quintet (with its minimal binding register defined as YCUNN), and this longer binding register, as compared to either RRM1 or RRM4 (discussed below), is facilitated by the extended β-sheet. While there is no aromatic residue at RNP2 position 2, as with all other RRMs of PTBP1, the C3 nucleotide still sits above this canonical position on the β-sheet of RRM3 (occupied in RRM3 by L366 [81]), and is coordinated by two hydrogen bonds from residues on the β4 strand (S435 [82] and K436) (Figure 4). The two uracil nucleotides flanking this C3 nucleotide (U2 and U4) are coordinated by a series of hydrogen bonds and hydrophobic interactions. U2 is H-bonded to T433, and sits above R431 on β4. U4 hydrogen bonds with β4-β5 loop residues H437 [82] and V440, and forms hydrophobic interactions with β3 residues K394, L396 [81] and L404 [81] (position 5 of RNP1) (Figure 4). Notably, these three nucleotides, which also make up the minimal binding register of RRM1, are coordinated on RRM3 in a similar manner as RRM1. The C-terminal C5 nucleotide is extensively coordinated, and its base interacts with L396 and stacks on F397, while its sugar interacts with L452, and its phosphate oxygen contacts K394 (Figure 4). A subsequent study demonstrated that mutation of K394, L396, and F397 significantly ablated RRM3 binding to ECMV and PV-1 IRESs, confirming the importance of these residues in driving target RNA binding [69]. U6 is less specifically coordinated [82], and is contacted by P443 and R444, while its phosphate oxygen also contacts K394 (Figure 4). Note that all residues are derived from Oberstrass et al. (2005) [31] and Auweter et al. (2007) [53], while additional references reflect subsequent studies confirming these interactions.

RRM4, like RRM1, binds the U2C3U4 triplet in a similar fashion but with several key differences (hence its minimal binding register—YCN—is slightly different from RRM1). Like RRM1, C3 stacks on H483 [82] (RNP2 position 2), and is H-bonded to two residues on the β4 strand (S553 [82] and K554 [82]) (Figure 4). A later study confirmed that mutation of S553 and K554 significantly reduced binding to ECMV and PV-1 IRESs, further confirming the importance of these residues in RNA binding [69]. However, in contrast with RRM1, U2 is not bound as tightly, and stacks and forms a hydrogen bond with the N474 side chain (interdomain linker) (Figure 4). Finally, the U4 nucleotide is coordinated in a hydrophobic pocket formed by L521 (RNP1 position 5), F513 and the aliphatic portions of K511 [81] and K515 [81] (Figure 4). Again, in contrast with RRM1, this U4 nucleotide is flipped 180 degrees, as the shorter C-terminal extension only allows one hydrogen bond with this nucleotide (I557 [81]), rather than the two seen in RRM1 (Figure 4). Intriguingly, a later study modeling changes in domain backbone dynamics upon formation of the RRM3-4 complex have indicated that there are changes in backbone dynamics upon association of RRM3 and RRM4 to form RRM3-4 (as compared to the individual domains) that could be important factors in ligand (e.g., RNA) selection [66]. Specifically, RRM3 appears to lose conformational flexibility upon association with RRM4.

The organization of these RRMs (juxtaposed so that their β-sheets face in opposing directions) is unique among RNA binding proteins with tandem RRMs, and has important functional implications. All other RNA binding proteins with tandem domains organize these domains such that they bind immediately adjacent stretches of nucleotides on target RNA [39,40,41]. The organization of RRM3-4 requires separation of the binding register for each domain by at least 15 nucleotides [31], thus RRM3-4 preferentially binds longer single-stranded RNA [67,70]. Importantly, this complex has been shown to facilitate induction of RNA looping [31,83], which is particularly relevant when considering PTBP1’s roles in alternative splicing and IRES-mediated translation, as both processes require RNA looping (either for exon exclusion or ribosomal binding, respectively).

### 2.5. Summary

In all RRMs of PTBP1, although there are substitutions of aromatic amino acids at key RNA interacting positions within RNP1 and RNP2, the residues at these conserved positions still contact target RNA as reflected in seminal studies mapping RNA: RRM interactions [31,53]. RNP2 position 2 and RNP1 position 5 of all four RRMs contact target RNA, as demonstrated in the studies referenced above [31,33,53,68,81,82]. Specificity of RNA: RRM interactions arise from the H-bond and hydrophobic pocket networks that coordinate each respective binding register, as well as the unique topologies between RRMs (extra β5 on RRM2 and RRM3) [29,35]. Each RRM has a different minimal binding register, different secondary structure preference and, in the case of RRM3-4, unique ability to manipulate the structure of RNA [31].

## 3. The RRM Domains of PTBP1 Have a Unique Contribution to PTBP1 Function

The idea of separation of function among the RRM domains of PTBP1 was proposed in the early PTBP1 literature, and has been refined by numerous subsequent studies over the roughly four decades since PTBP1’s initial discovery. These early PTBP1 studies had suggested that RRM3 and RRM4 (RRM3-4) were the major drivers of RNA binding (particularly RRM3 [28,32]), while RRM1 and RRM2 served as protein interacting domains and had little role in RNA binding [23,28,84]. Indeed, Oh et al. had designated RRM1 as a HeLa factor responding domain, for its apparent ability to enhance RNA binding through interaction with HeLa cell cytoplasmic factors [84]. This study and another [28] also suggested that RRM2 drove PTBP1 dimerization rather than RNA binding—a phenomenon that was later found to be an artifact of the size exclusion chromatography methodology [30,52,85,86].

Subsequent biochemical studies revealed that all four RRMs of PTBP1 bind RNA, demonstrating that the separation of functions within this protein is not as clearly delineated as “protein interacting” domains and “RNA binding” domains. Instead, these studies suggest that separation of functions within the PTBP1 protein is a result of unique sequence and structure preferences of each RRM for target RNA. While all four RRMs of PTBP1 utilize the residues at RNP2 position 2 (or in the case of RRM2, the canonical position) and RNP1 position 5 to coordinate RNA, the network of hydrogen bonds and hydrophobic interactions that drive the specificity of RRM: RNA interactions are unique between the RRMs. Therefore, as described in the previous sections of this review, each RRM has a unique binding register, secondary structure preference and, especially in the case of RRM3-4, the ability to manipulate the structure of target RNA. This, combined with the solution behavior of the full PTBP1 protein (RRM1 and RRM2 move independently in solution due to long flexible C-terminal linker regions, whereas RRM3-4 is in a constitutive didomain conformation), supports the idea that each RRM may provide unique contributions to independently modulate RNA specificity and, ultimately, PTBP1 function. In this section, we summarize the current evidence for domain-specific functions of each of PTBP1’s RRM domains (or didomain in the case of RRM3-4).

### 3.1. RNA Recognition Motif 1 (RRM1)

Of the four RRMs of PTBP1, RRM1 has the most numerous examples in the literature of discrete domain-specific functions. As described above, a C-terminal α3 helix folds upon binding to an RNA hairpin in the ECMV IRES, and this helix was critical for PTBP1-mediated enhancement of IRES activity in vitro [68]. While this has not yet been explored in vivo, this study and another [72] have revealed a potential mechanism of domain-specific RRM1 function in PTBP1-dependent IRES initiated translation, the functional implications of which are important to elucidate in an in vivo setting.

It is known that PTBP1 regulates multiple aspects of the biogenesis of several apoptotic genes [87,88,89,90]. RRM1, specifically, has been shown to be important for maintaining the expression of Caspase-9, BAX, and BID in differentiating cardiomyocytes, as deletion of this domain resulted in a decreased expression of these transcripts. Zhang and colleagues further demonstrated that RRM1 plays a role in regulating exon 10 inclusion in its homolog PTBP2, as truncation of the protein to exclude RRM1 and the N-terminal NLS resulted in increased exon 10 skipping in PTBP2 [91]. This is particularly biologically relevant given that this is the mechanism by which PTBP1 represses PTBP2 signaling: exon 10 skipping leading to nonsense mediated decay of the PTBP2 transcript [92].

Finally, RRM1 has also been shown to have a critical function in regulating genes that control invasive potential in cancer. A study by Wang et al. demonstrated that PTBP1 binds the 5′ UTR (untranslated region) of the hypoxia inducible factor 1α (HIF-1α) transcript via its RRM1 and RRM3 domains, destabilizing HIF-1α and promoting an invasive phenotype in a non-small cell lung cancer (NSCLC) model [93]. More recently, this group also demonstrated that PTBP1 negatively regulates the AXL tyrosine kinase transcript by binding its 5′ UTR, resulting in reduced invasive potential, and that RRM1 is critical for this binding interaction [94]. Although RRM1 has conflicting effects on invasive potential in these two model systems, these studies demonstrate that RRM1 can impact cellular phenotype through regulation of distinct RNA targets. Importantly, supplementing this growing literature, we have recently shown that RRM1 contains a reverse Bcl-2 homology domain 3 (rBH3) motif regulatory sequence. This sequence allows the anti-apoptotic protein, MCL1, to displace target RNA from RRM1, establishing a mechanism by which these emerging functions of RRM1 can be independently regulated [95].

### 3.2. RNA Recognition Motif 2 (RRM2)

It has been well described that RRMs can serve as protein–protein interaction interfaces in addition to their canonical RNA binding function [35,62,96]. In the early PTBP1 literature, RRM2 was initially thought to be a protein interacting domain of PTBP1, with a specific role in mediating oligomerization of PTBP1 [28,84]. Although later studies revealed that PTBP1 was a monomer in solution and did not dimerize via its RRM2 domain, the concept that RRM2 can serve as a protein–protein interaction interface in addition to its RNA binding properties was demonstrated in subsequent studies. Raver1 was initially identified as a PTBP1 binding partner via yeast two-hybrid screen [97], and it was later shown that this interaction with PTBP1 was necessary for repression of exon 3 in α-tropomyosin [98]. Subsequent studies by the same group identified that RRM2 and the adjacent linker region were the minimal repressor domain for α-tropomyosin exon 3 repression [99], and that Raver1 interacts with a hydrophobic groove on the α-helical surface of RRM2 via a defined peptide motif ([S/G][I/L]LGxPP) [100]. Of note, this motif is also present on Raver2 and Matrin3, suggesting that these proteins could also play a role in regulation of other RRM2-dependent functions [100,101]. Additionally, Kafasla et al. demonstrated that deletion of RRM2 ablated ECMV IRES activity (but not poliovirus IRES activity), indicating that interrogation into impact of RRM2 protein–protein interactions on IRES function is necessary [70].

### 3.3. RNA Recognition Motifs 3 and 4 (RRM3-4)

While the three isoforms of PTBP1 formed by alternative splicing at exon 9 are well described, there is also a fourth PTBP1 isoform—formed by the removal of exons 2-10—that contains only RRM3 and RRM4 [1,3,5,75,102]. The presence of an endogenous independent RRM3-4 didomain suggests these two RRMs have biologically relevant independent functions. Both the hepatitis A virus [103] and poliovirus 3C [104] proteinases cleave PTBP1 at the linker region between RRM2 and RRM3 to release the RRM3-4 didomain, further underscoring the independent biological function of this didomain.

As described above, the geometric orientation of these domains (in which their RNA binding interfaces face in opposite directions) is well poised to induce RNA looping and, indeed, prior studies have confirmed this ability of the RRM3-4 didomain [31,83]. It has been suggested that this RNA looping function is important in processes such as alternative splicing (looping out of a repressed exon) and IRES-mediated translation (looping to create a ribosomal binding site). Accordingly, several studies have demonstrated a role of this didomain in regulation of alternative splicing. One study revealed that RRM4 was required for switching from a neural to non-neural splicing pattern in a C28 splicing assay, as mutation of several key RNA-interacting residues of RRM4 uncoupled RNA binding from splicing activity [105]. In another study, RRM3 and RRM4 were found to be required to mediate exon 11 repression of PTBP1 itself in a negative feedback loop, as a truncated protein expressing just the RRM1 and RRM2 domains failed to repress exon 11 [106]. A similarly truncated PTBP1 protein (containing only RRMs 3 and 4) was also found to have reduced splicing capacity of a cardiac troponin exon 5 minigene reporter in vivo [107]. Finally, RRM3-4 function was found to be essential for splicing repression in a splicing reporter assay (using the DS9-175 minigene) in vivo, and this was likely due to its ability to induce RNA looping [83].

## 4. Perspectives and Conclusions

Since its initial identification in 1988, PTBP1 has taken a center stage as a model protein for studying RBP biology and function. Its roles in virtually all stages of RNA maturation and utilization (e.g., 3′ and 5′ end processing, splicing, IRES translation) have provided a rich substrate for study, as by these ubiquitous functions PTBP1 impacts a wider range of biological processes than any other RBP. It has been described as a critical regulator of gene expression homeostasis in various tissues, has been implicated in regulating development in multiple tissue types, and its dysregulation has been shown to drive phenotypes in multiple disease states including, but not limited to, various cancers.

Despite PTBP1’s position as a main character in the field of RBP biology, there has been little description of the dynamic anatomy of this protein and how its various components (here, individual RRMs) coordinate its diverse functions. In the early PTBP1 literature, the available genomic techniques did not allow for high-resolution and throughput analyses in cellular-based assays to elucidate PTBP1’s intra-protein functional dynamics in an endogenous environment. However, recent advances in genomic technology now provide a toolbox to answer these previously unanswered questions. The advent of CRISPR-based technology now allows the capability of deleting portions of or entire protein domains to interrogate their biological function endogenously. More efficient and cost-efficient immunoprecipitation and sequencing techniques allow higher resolution mapping of binding sites and even interrogation of RNA structure, when it is bound to proteins of interest. These approaches allow us the enhanced molecular resolution necessary to begin mapping modular roles of individual domains of RBPs on a target-specific basis.

The current lack of knowledge of the dynamics of RBP—and specifically PTBP1—coordination of functions has rendered these proteins essentially “undruggable” and, therefore, most effort in therapeutic development has focused on interventions downstream of RBPs—e.g., targeting the dysregulated RNA or protein product. Deconstructing the modularity of RBPs will allow more targeted upstream interventions of dysregulated RBP function. For example, if a particular domain of an RBP drives a pathologic splicing event—e.g., RRM2 and α-tropomyosin—a small molecule inhibitor can be designed for the specific ligand (RNA): RRM that does not impact the function of the RBP at other targets. Our goal is that the information contained in this review will frame the structure–function context of each RRM of PTBP1 and facilitate the next generation of PTBP1 interrogation—moving from studying the macroscopic cellular function of the entire protein, to a higher-resolution view of how the protein coordinates these observed functions. Ultimately, we hope that this will provide a foundation for understanding how individual RBD domains (and their specific RNA interactions) can be targeted to treat developmental defects and disease.

## Figures and Tables

**Figure 1 ijms-24-11218-f001:**
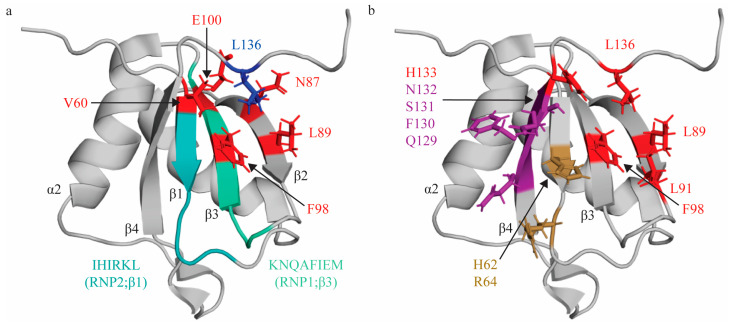
Topologic organization and key features of RRM1. (**a**) Apo structure of RRM1 (PDB: 1SJQ [30]) with residues comprising the RNP1 sequence on β3 shaded green (K94, N95, Q96, A97, F98, I99, E100, M101) and residues of RNP2 on β1 shaded in teal (I61, H62, I63, R64, K65, L66). C terminal residue (L136; blue) is stabilized by hydrophobic contacts with several residues across the β-sheet (V60, L89, F98, N87, E100; red). (**b**) Key residues that interact with the minimal RNA binding register YCN. Residues that make stacking interactions (H62, R64) are colored sand yellow. Residues that make hydrogen bonds (Q129, F130, S131, N132) are colored purple. Note that N132 also makes a stacking interaction with the C_3_ nucleotide, but is colored based on hydrogen bond in this figure. Residues that engage in hydrophobic interactions (L89, L91, F98, H133, L136) are colored red.

**Figure 2 ijms-24-11218-f002:**
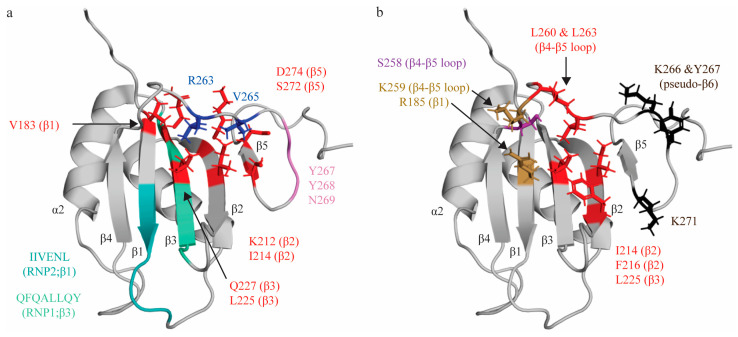
Topologic organization and key features of RRM2. (**a**) Apo structure of RRM2 (PDB: 1SJR [30]) with residues comprising the RNP1 sequence on β3 shaded green (Q221, F222, Q223, A224, L225, L226, Q227, Y228) and residues of RNP2 on β1 shaded teal (I186, I187, V188, E189, N190, L191). C-terminal residues (R263, V265; blue) are stabilized by hydrophobic contacts with several residues across the β-sheet (V183, I214, L225, K212, Q227, S272, D274; red). Residues Y267, Y268 and N269 for a pseudo-β6 strand (pink). (**b**) Key residues that interact with the minimal RNA binding register CU(N)N. Residues that make stacking interactions (R185, K259) are colored in sand yellow. Residues that make hydrogen bonds (S258) are colored purple. Note that the main chain of K259 also forms an H-bond, but is colored based on stacking interaction in this figure. Residues that engage in hydrophobic interactions (I214, F216, L225, L260, L263) are colored red. Residues with undefined contacts with RNA (K66, Y267, K271) are colored black.

**Figure 3 ijms-24-11218-f003:**
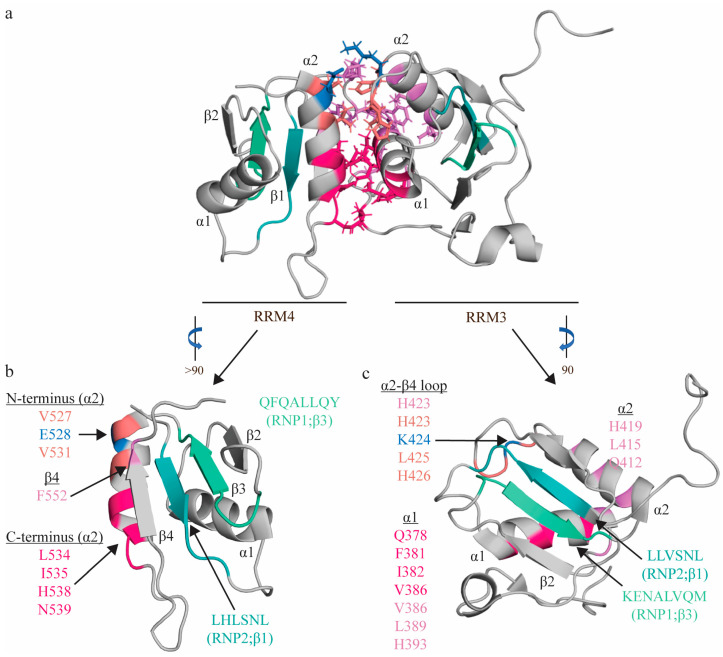
Topologic organization and key features of apo RRM3-4. Apo structure of the RRM3-4 didomain (PDB: 2EVZ [33]) (**a**), with individual views of RRM4 (**b**) and RRM3 (**c**). Residues comprising the RNP1 sequences on β3 shaded green (RRM3: K400, E401, N402, A403, L404, V405, Q406, M407; RRM4: R517, K518, M19, A520, L521, I522, Q523, M524) and residues of RNP2 on β1 shaded teal (RRM3: L365, L366, V367, S368, N369, L370; RRM4: L482, H483, L484, S485, N486, L487). The C-terminal region of RRM4 α2 (L534, I535, H538, N539; hot pink) interacts with L461 (hot pink) of the interdomain linker, as well as with the N-terminal region of RRM3 α1 (Q378, F381, I382, V386; hot pink), positioning these helices perpendicularly to one another. The N-terminal region of RRM4 α2 (V527 (salmon), E528 (blue), V531 (salmon)) interacts with the α2-β4 loop of RRM3 (H423 (salmon), K424 (blue), L425 (salmon), H426 (salmon)). Most contacts between the interdomain linker (F464, F472, I475, P477, and P478; not individually identified in this figure) are between α1 (V386, Y387, D389; pink) and α2 of RRM3 (Q412, L415, H419; purple), along with H423 (salmon) of the following loop, with an additional contact with RRM4 (F552 of β4 of RRM4; purple). Finally, this interaction is stabilized by an ion pair (K424 on RRM3 and E528 on RRM4; blue). Note that colors correspond to residues that interact with one another, and residues with multiple contacts (V386, H423) are reported twice in the appropriate colors.

**Figure 4 ijms-24-11218-f004:**
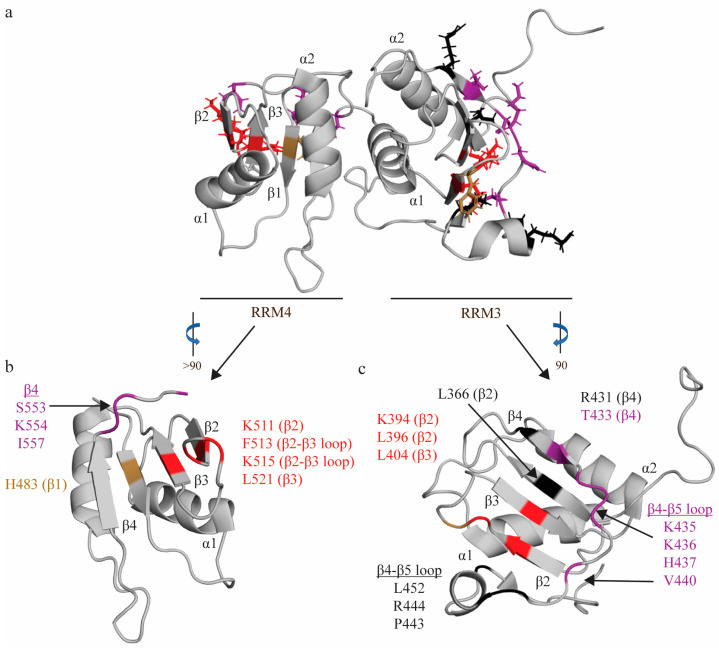
RNA interactions of RRM3-4. (**a**) Structure of RRM3-4 didomain (PDB: 2EVZ [33]) with key residues that interact with the minimal binding register of RRM4 (YCN) and RRM3 (YCUUN), highlighted in colors that reflect the nature of their chemical interaction with RNA nucleotides. Residues that make stacking interactions are colored sand yellow, residues that engage in hydrophobic interactions are colored red, and residues that make hydrogen bonds are colored purple (see descriptions below). (**b**) Key residues of RRM4 that interact with the minimal RNA binding register (YCN). Residue H483 makes a stacking interaction. Residues N448, S553, K554, I557 make hydrogen bonds. Residues K511, F513, K515, L521 engage in hydrophobic interactions. (**c**) Key residues of RRM3 that interact with the minimal RNA binding register (YCUNN). Residue F397 makes a stacking interaction. Residues T433, S435, K436, H437, V440 make hydrogen bonds. Residues K394, L396, L404 engage in hydrophobic interactions. Residues L366—RNP2 position 2, L396, R431, L452, P443, and R444 have undefined contacts with RNA and are colored black.

**Table 1 ijms-24-11218-t001:** RNP1 and RNP2 sequences for all four RRMs of PTBP1. Conserved aromatic residues in the consensus and PTBP1 RRM sequences are shown in red. Substitutions of these residues in each PTBP1 RRM are in blue. Note that the RNP2 sequence of RRM2 (IIVENL) is shifted by two residues in RRM2′s structure due to an insertion in the β1 strand of RRM2.

	RNP2	RNP1
**Position**	** 1 2 3 4 5 6 **	** 1 2 3 4 5 6 7 8 **
**Consensus sequence**	** L F V G N L ** ** I Y I K L L **	** K G F G F V X F ** ** R G Y A F V X Y **
**RRM1 sequence**	** I H I R K L **	** K N Q A F I E M **
**RRM2 sequence**	** L R I I V E N L **	** Q F Q A L L Q Y **
**RRM3 sequence**	** L L V S N L **	** K E N A L Q V M **
**RRM4 sequence**	** L H L S N L **	** R K M A L I Q M **

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
