# Peer review of "Reviewing PTBP1 Domain Modularity in the Pre-Genomic Era: A Foundation to Guide the Next Generation of Exploring PTBP1 Structure–Function Relationships"

_ijms, 2023, doi:10.3390/ijms241311218_

Round 1

Reviewer 1 Report

The manuscript is a review of the role of different domains of PTB1. The authors have done an extensive job to review the detailed structure of each domain and then connecting these different domains to the function of the protein.

Below are some suggestions that can improve the content and scope of the review.

1.  The major point: The review lacks an introduction to the family of PTB proteins. Particularly the role and function of the tissue specific expression of neuronal PTB (also known as PTB2) should be discussed.

2. The title is a bit misleading as the review's major focus is on describing the structure of different domains with in PTB1. The review's current title does not reflect the content. The provided review is very limited to the discussion of structural components.

2. The authors should review the document for typos and also ensure that each abbreviation is expanded at its first mentions. For e.g. hnRNP in line 28.

3. The resolution of the figures is very poor. Please use high resolution figures.

4. No information is provided on the PDB ID of the structures used in the figures.

5. The conclusion section can be improved further to include more functional and future studies.

Acceptable, a general review will be helpful

Author Response

Response Reviewer #1:

The manuscript is a review of the role of different domains of PTB1. The authors have done an extensive job to review the detailed structure of each domain and then connecting these different domains to the function of the protein.

Below are some suggestions that can improve the content and scope of the review.

1.  The major point: The review lacks an introduction to the family of PTB proteins. Particularly the role and function of the tissue specific expression of neuronal PTB (also known as PTB2) should be discussed.

- We agree that a more detailed analysis on the interplay between PTB1 and PTB2 could be written, but we lack space to clearly outline the nuanced details of this. We have added a set of sentences highlighting this interplay in the introduction. (Page 1: lines 40-43)

2. The title is a bit misleading as the review's major focus is on describing the structure of different domains with in PTB1. The review's current title does not reflect the content. The provided review is very limited to the discussion of structural components.

- the focus is on the structure-function of the domains, we have changed the title to highlight the domain focus on the review and part of the goal is that this review serves as a cap on the structure aspect and an introduction to the function aspect of the study of PTBP1

2. The authors should review the document for typos and also ensure that each abbreviation is expanded at its first mentions. For e.g. hnRNP in line 28.

- thank you for pointing out this oversight on our part, we have reviewed and believe we have expanded all abbreviations at first mentions

3. The resolution of the figures is very poor. Please use high resolution figures.

- we have been in communication with the editors and are uploading .tiff files of all figures that should address quality of the figures

4. No information is provided on the PDB ID of the structures used in the figures.

- we apologize for this oversight and have added the PDB ID used as the basis for all figures to each figure caption.

5. The conclusion section can be improved further to include more functional and future studies.

- we believe the review is already quite long and proposing the wide range of functional and future studies that are needed would be both a large expanse of the text and perhaps limiting to our vision

Reviewer 2 Report

The topic of the manuscript under review is interesting, the authors attempt to discuss in detail the issues related to the well-known protein PTBP1, its domains and interactions with RNA, and finally to find connections between its various catalytic functions.  To some extent, they have succeeded in doing so. However, some parts are written in a well-legible style, while others are just the opposite. From the point of view of my expertise, I have no serious comments to make. The authors often use a number of not-so-common abbreviations in the text, so it would be useful to include a list of abbreviations in this manuscript.
In Table 1 there are asterisks, however, there is no explanation of what they are meant to  emphasize.
The quality of images in Figures 1-4 is very low, it is certainly necessary to increase the resolution, which makes their assessment more difficult. These surely require to be re-made.
The RRM-RNA complex is better written with a hyphen than with a colon. The abbreviation  PTBP1:34 is, in my opinion, rather confusing. 

Author Response

Response Reviewer #2:

The topic of the manuscript under review is interesting, the authors attempt to discuss in detail the issues related to the well-known protein PTBP1, its domains and interactions with RNA, and finally to find connections between its various catalytic functions.  To some extent, they have succeeded in doing so. However, some parts are written in a well-legible style, while others are just the opposite. From the point of view of my expertise, I have no serious comments to make. The authors often use a number of not-so-common abbreviations in the text, so it would be useful to include a list of abbreviations in this manuscript.

- We have gone through and made sure to expand on all abbreviations

In Table 1 there are asterisks, however, there is no explanation of what they are meant to  emphasize.

- The asterisks were initially meant to bring attention to the shift in the sequence for RRM2, but upon further review this was not felt to be necessary outside of the note in the table caption.

The quality of images in Figures 1-4 is very low, it is certainly necessary to increase the resolution, which makes their assessment more difficult. These surely require to be re-made.

- we have been in communication with the editors and are uploading .tiff files of all figures that should address quality of the figures

The RRM-RNA complex is better written with a hyphen than with a colon. The abbreviation  PTBP1:34 is, in my opinion, rather confusing. 

- we recognize that the colon abbreviation did not work as well as intended given the density of information in the review. We looked at a hypen abbreviation (PTBP1-34) and felt that this also was confusing. After further review, we have chosen to go forward with a subscript abbreviation as this is used often in other manuscripts to signify a truncation, domain, or mutant of a protein (PTBP1RRM3-4). We hope that this improves readability of the overall manuscript.

Reviewer 3 Report

In this Review, the authors illustrate the biochemical structure of the RNA-binding protein PTBP1 in order to pave the way for future research. The PTBP1 protein is fully and extensively described, highlighting the relevance of several protein motifs and residues to define the protein structure and function. There is exhaustive research to portray the biochemical nature of PTBP1. Yet, the text somehow fails to pinpoint the biological importance of studying PTBP1. In addition, I find that the text lacks appeal to a broad audience. In my opinion, several points should be addressed before publication.

Major concerns:

1-PTBP1 is present in all organisms or is it restricted to animals? Its well-known functions are conserved across kingdoms? This important piece of information is absent (i.e. Abstract, Intro).

2-Is the heterogeneous structure of the PTBP1 conserved? Is it always have four RRM motifs? What is the level of conservation of the RRM motifs between different species?

3-The sequence of Table 1 must be aligned to show the substitutions and the insertion in RRM2 (and the numbers of positions should be corrected accordingly).

4-The first paragraph of Perspective and Conclusions (L467-474) should be relocated to a different section (Intro maybe). It does not say anything about future directions. Instead, it might be used to portray PTBP1 role and relevance at the beginning of the Review.

5-In line with my previous comment, I think that both the biological and molecular role of PTBP1 should be better illustrated.

6-What are the most relevant motifs to explore in the future? Considering the information gathered in the text.

7-In general, the text is plain. Moreover, there are several grammar mistakes that needs further attention.

8-At what extent PTBP1 impacts alternative splicing? Is there any preference for different splicing events (i.e.: intron retention, etc.)?

Minor concerns:

Abstract

L10: I am not familiarized with the concept of RNA utilization. I work with plant RNA-binding proteins and alternative splicing. Can you expand the meaning of such a concept for a more general audience?

L13: I suggest to rephrase “Unfortunately”.

Intro:

L34: “affinity directly correlated” missing verb.

L61: missing verb

I marked some mistakes, although I am not a native speaker.

Author Response

Response Reviewer #3:

In this Review, the authors illustrate the biochemical structure of the RNA-binding protein PTBP1 in order to pave the way for future research. The PTBP1 protein is fully and extensively described, highlighting the relevance of several protein motifs and residues to define the protein structure and function. There is exhaustive research to portray the biochemical nature of PTBP1. Yet, the text somehow fails to pinpoint the biological importance of studying PTBP1. In addition, I find that the text lacks appeal to a broad audience. In my opinion, several points should be addressed before publication.

Major concerns:

1-PTBP1 is present in all organisms or is it restricted to animals? Its well-known functions are conserved across kingdoms? This important piece of information is absent (i.e. Abstract, Intro).

  • This would be a huge discussion in and of itself. We believe that a phylogenetic analysis of PTBP1 and its domains is well outside the scope of the current review, but that such a study would be of significant interest to the field.

2-Is the heterogeneous structure of the PTBP1 conserved? Is it always have four RRM motifs? What is the level of conservation of the RRM motifs between different species?

  • See response to 1, above

3-The sequence of Table 1 must be aligned to show the substitutions and the insertion in RRM2 (and the numbers of positions should be corrected accordingly).

  • The sequences in Table 1 do not have any substitutions and or insertions in RRM2, the only point is that the sequence is shifted in the structure. This has been clarified in the Table Caption

4-The first paragraph of Perspective and Conclusions (L467-474) should be relocated to a different section (Intro maybe). It does not say anything about future directions. Instead, it might be used to portray PTBP1 role and relevance at the beginning of the Review.

  • This first paragraph is meant to reframe the manuscript and provide a conclusion so that the subsequent future directions can be easily interpreted.

5-In line with my previous comment, I think that both the biological and molecular role of PTBP1 should be better illustrated.

  • The goal of this review is to highlight that the domain-dependent molecular role of PTBP1 has been significantly understudied.

6-What are the most relevant motifs to explore in the future? Considering the information gathered in the text.

  • At present, sequence motifs in RNA remain those previously identified in the studies cited. We hope that the review makes it clear that other motifs may arise, but we do not have information on which should be immediately targeted.

7-In general, the text is plain. Moreover, there are several grammar mistakes that needs further attention.

  • Our goal is to have a direct and clear discussion of a complex set of both structural and biochemical data. We apologize for the grammatical mistakes and believe we have addressed most of these. We apologize for missing these during our preparation of the manuscript.

8-At what extent PTBP1 impacts alternative splicing? Is there any preference for different splicing events (i.e.: intron retention, etc.)?

  • PTBP1 has been shown to be involved in both intron excision, exon inclusion or retention, and alternative 5’ or 3’ start/end

Minor concerns:

Abstract

L10: I am not familiarized with the concept of RNA utilization. I work with plant RNA-binding proteins and alternative splicing. Can you expand the meaning of such a concept for a more general audience?

  • We attempted to combine PTBP1’s impact on RNA stability and translation, but agree that this ultimately was confusing and have changed this

L13: I suggest to rephrase “Unfortunately”.

  • We have changed this

Intro:

L34: “affinity directly correlated” missing verb.

  • In this instance, correlated is the verb

L61: missing verb

  • corrected

Round 2

Reviewer 2 Report

The revised version is now ready for acceptance.

Reviewer 3 Report

The authors have addressed all the comments. The work is now ready for publication.

The text has been improved.